# Fundamentals of Plant Morphology and Plant Evo-Devo (Evolutionary Developmental Morphology)

**DOI:** 10.3390/plants12010118

**Published:** 2022-12-26

**Authors:** Rolf Sattler, Rolf Rutishauser

**Affiliations:** 1Biology Department, McGill University, Montreal, QC H3G 0B1, Canada; 2Department of Systematic and Evolutionary Botany, University of Zurich, CH-8008 Zurich, Switzerland

**Keywords:** classical morphology, continuum morphology, process morphology, fuzzy logic, homeosis, plant evo-devo, plant molecular genetics

## Abstract

Morphological concepts are used in plant evo-devo (evolutionary developmental biology) and other disciplines of plant biology, and therefore plant morphology is relevant to all of these disciplines. Many plant biologists still rely on classical morphology, according to which there are only three mutually exclusive organ categories in vascular plants such as flowering plants: root, stem (caulome), and leaf (phyllome). Continuum morphology recognizes a continuum between these organ categories. Instead of Aristotelian identity and either/or logic, it is based on fuzzy logic, according to which membership in a category is a matter of degree. Hence, an organ in flowering plants may be a root, stem, or leaf to some degree. Homology then also becomes a matter of degree. Process morphology supersedes structure/process dualism. Hence, structures do not have processes, they are processes, which means they are process combinations. These process combinations may change during ontogeny and phylogeny. Although classical morphology on the one hand and continuum and process morphology on the other use different kinds of logic, they can be considered complementary and thus together they present a more inclusive picture of the diversity of plant form than any one of the three alone. However, continuum and process morphology are more comprehensive than classical morphology. Insights gained from continuum and process morphology can inspire research in plant morphology and plant evo-devo, especially MorphoEvoDevo.

## 1. Introduction

Traditionally, plant morphology has been one of the major disciplines of plant biology or botany besides plant physiology, genetics, systematics, ecology, and evolution. However, more recently, plant morphology has become absorbed to a great extent into plant evo-devo (evolutionary developmental biology) as a subdiscipline. In fact, evo-devo comprises morphology and molecular genetics [1,2]. Wanninger [3] emphasized the importance of morphology in evo-devo and referred to “MorphoEvoDevo”, that is, evolutionary developmental morphology.

To describe and explain the enormous diversity of plant form, including the development and evolution of plant form, we need concepts and a conceptual framework that are fundamental in any discipline. In this article, we shall discuss the concepts and conceptual framework of classical morphology, continuum morphology, and process morphology and point out how they influence research in plant morphology and plant evo-devo. Since morphological concepts such as root, stem or leaf are also used in other botanical disciplines such as plant physiology, genetics, and systematics, our discussion will also be relevant to these disciplines [4].

## 2. Classical Morphology

Classical plant morphology (typology) with sharp and rigid definitions of structural categories has one of its roots in Goethe’s *Metamorphosis of Plants* [5]. Goethe was a classical and romantic poet and scientist who coined the term “morphology.” In his booklet *The Metamorphosis of Plants*, he postulated that flowering plants consist of three fundamental kinds of organs: root, stem, and leaf, and that all lateral appendages from the cotyledons to the foliage leaves to the organs of the flower are all “one and the same” organ, which means that they share the same essence, although they may appear very different in their morphology [6,7]. This idea of essentialism is still alive in mainstream morphology up to the present time: there are only three organ categories in vascular plants such as flowering plants that are mutually exclusive because they have different essences. Although most modern plant morphologists may not refer to essences, they insist that any organ must be either a root, a stem, or a leaf homologue. The general term “caulome” is used for stems and the term “phyllome” for all leaf homologues. The shoot comprises both caulomes and phyllomes. In this sense, the most comprehensive treatise of plant morphology in the 21st century, “Kaplan’s Principles of Plant Morphology” [8], appears fundamentally classical, thus subscribing to the root, caulome, and phyllome trinity of organs in vascular plants such as flowering plants [9]. 

Although Goethe proposed this trinity, he also entertained other rather divergent views on the morphology of plants. In his *Metamorphosis of Plants* (1790), he also subdivided plants into only two units: roots and phytomeres [5,6,7]. A phytomere consists of a leaf, its axillary bud, the node and one internode below it. It underlines the “stem-node-leaf continuum” [10]. In other writings, Goethe proposed that “all is leaf,” and he even anticipated Agnes Arber’s [11] partial-shoot theory of the leaf when he wrote: “When leaves divide, or rather when they advance from their original state to diversity, they are striving toward greater perfection, in the sense that each leaf has the intention of becoming a branch” [7]. These views enlarge the scope of morphology enormously but usually are ignored. 

However, Goethe’s morphology was pre-Darwinian and angiosperm-centric. After Darwin, many morphologists understood Goethe’s concepts in an evolutionary perspective and applied them beyond angiosperms. However, often they retained the same concepts [8]. Others have proposed alternative approaches, which have not been incorporated into mainstream morphology (e.g., [11]) because mainstream morphology appears to be fundamentally wedded to Aristotelian either/or logic, according to which any organ must be either a root, a caulome, or a phyllome. Since there are organs that do not fit into these categories, endless debates ensued about whether they are essentially a root, a caulome, or a phyllome. These debates appear futile because they are based on either/or logic that cannot resolve the issues.

Bell [12] referred to structures that do not fit into the classical categories as “morphological misfits,” and he stressed that they are only “misfits to a botanical discipline [such as classical morphology], not misfits for a successful existence” [12]. “Various morphological misfits emerged as morphological key innovations (perhaps ‘hopeful monsters’) that gave rise to new evolutionary lines of organisms” [2]. An example is the novel pathways in *Utricularia* (Lentibulariaceae) (see below). 

Morphological misfits have been said to have “identity crises” [13]. “Multicellular plants such as angiosperms are used to having identity crises on various levels, from cells to meristems and organs and even beyond. Identity crises, however, are not the problem of the plants, but of our inadequate thinking and concepts” [13] (p. 196). Concerning the genetic basis, Vergara-Silva [14] (p. 260) noted: “Distinct groups of genes that in principle act in one categorical structure, are actually also expressed in another, and…the consequence that this overlapping pattern has on cell differentiation is an effective blurring of the phenotypic boundary between the structures themselves”.

Nonetheless, classical morphologists forced all structures into their categorical framework, which has led to almost endless controversy about structures that do not fit into the categories. For example, the shoots of *Utricularia purpurea* … were interpreted as having no leaves at all [15]. Troll and Dietz [16] concluded similarly with respect to the shoots of terrestrial and epiphytic *Utricularia* species. According to them, the so-called ‘leaves’ of *U. longifolia* are nothing but ‘phylloclades’, i.e., phyllomorphic shoots. Kaplan [8] (p.570) came to the opposite conclusion with respect to the stolons of, e.g., *Utricularia alpina*. He wrote: “… we interpret this system of axes and their branches in *U. alpina* as leaf homologues.” According to him it is the “leaf-borne shoot which ultimately elongates into the scapose inflorescence”. 

## 3. Continuum Morphology

In contrast to classical morphology, continuum morphology is not exclusively based on either/or logic but rather on a continuum logic that has become known as “fuzzy logic” [17]. Accordingly, categories such as stem and leaf (in vascular plants such as flowering plants) become fuzzy sets. Membership in fuzzy sets ranges from 0% to 100%. 0% means that the structure is not a member of the set; 100% means that it is a typical member of the set. For example, a typical stem is a 100% or near 100% member of the set “stem.” Structures that deviate more or less from 100% have a value somewhere between the extremes of 0% and 100%, forming a continuum between the categories of classical plant morphology [18]. 

In a way, it seems unfortunate that the founder of fuzzy set theory chose to call it by that name. He could have called it continuum theory. For most people, including most biologists, the word ‘fuzzy’ is associated with vagueness and imprecision. However, fuzzy logic is much more precise than Aristotelian either/or logic because fuzzy logic implies a semi-quantitative or quantitative description, whereas either/or logic forces the whole range of forms into the two extremes of 0% and 100%. Then, everything between these two extremes is lost or distorted. The behavioral biologist Bernard Hassenstein pointed out that life must be seen as “injunction,”, i.e., as a concept that cannot be defined by a clear-cut set of properties [19]. Illustrating the problem, Hassenstein asked the question “How many grains result in a heap?” There is no clear-cut answer to this question. It depends on our perspective (including the size of the grains), if five, 20, or 50 grains are needed as a minimum to get a heap. 

What is the empirical evidence for a continuum of plant organs as constituted by fuzzy logic? There are at least three different areas of evidence: morphological, mathematical, and molecular genetics. In the 19th, 20th, and 21st centuries, plant morphologists have provided much evidence for a continuum between the structural categories (see, for example, [2,11,20,21]). Despite this evidence, plant morphology remained classical to a considerable degree. Thinking in terms of either/or appears easier. Fuzziness, despite the evidence, is not appreciated by the majority of plant morphologists and other biologists.

However, in addition to morphological evidence, mathematical analyses also support the continuum view of plant form. Through multivariate analysis such as principal components analysis, a morphospace based on a variety of morphological parameters was constructed that made it possible to calculate the morphological distance between plant organs and other structures [18,22]. For example, multivariate analysis has shown that the stamens of the parasitic plant *Comandra umbellata* (Santalaceae) are 51% phyllomic and 49% caulomic [18] (p. 261). Other mathematical approaches also confirm the continuum between structural categories in flowering plants [23].

Finally, investigations in plant molecular genetics agree to a considerable degree with the continuum view [24]. For example, a genetic basis has been demonstrated for the continuum between radial stems and dorsiventral leaves in flowering plants: “It is now widely accepted that … radiality and dorsiventrality are but extremes of a continuous spectrum. In fact, it is simply the timing of the KNOX gene expression.” [25] (p. 17). However, KNOX genes have also a plethora of functions predating land plants [26,27]. 

One reason why continuum morphology has not been widely accepted in botany is because of the emphasis that has been given to the position criterion in establishing homology. Even when structures such as leaves or roots had multiple evolutionary origins [28] and thus are defined within a specific clade such as in taxic homology [29], position still plays an important role [8,30,31]. Hence, it is often thought that—in most vascular plants—the position of an organ determines its homology [8]. According to Kaplan, “organs in plants are defined principally by their topographic-positional relationships [8] (p. 265). For example, a leaf is defined as a lateral organ subtending an axillary branch. Other classical morphologists also supported the prominence of the position criterion [32,33,34]. This kind of reasoning has, however, been undermined by the phenomenon of homeosis. Homeosis is the replacement of one structure or a trait of one structure by another one [35]. For example, in the phylloclades of some genera of the Asparagaceae, axillary branches (shoots) have been replaced by leaves or intermediates between a leaf and a branch [36]. The replacement is recognized by the special quality of the phylloclades, that is, the quality criterion of homology, not by the position criterion. The quality criterion “refers to characteristics of an organ which are distinctive” [8]. 

In addition to the morphological quality, the homeotic interpretation of the phylloclades is also supported by investigations of molecular genetics [37,38]. Thus, in the phylloclades of *Ruscus aculeatus* (Asparagaceae) genes of both the shoot apex and leaves are expressed. Hence, “the phylloclade is not homologous to either the shoot or the leaf, but it has a double identity” [37]. Aristotelian either/or logic is therefore not appropriate.

Heterotopy, the change in the position of structures (positional shifting), also undermines the general validity of the position criterion [39,40]. For example, in *Nasturtium officinale* (Brassicaceae) roots, like lateral shoot buds, are formed exogenously in the axil of leaves [41]. These roots are recognized as roots by their special quality, not by their position, which again underlines the importance of the quality criterion for homologization. 

Besides the position and quality criteria of homology, the transition criterion has also been used. According to this criterion, two structures are homologous if they are linked through intermediate transitional forms. A well-known example is the stamens of *Nymphaea* (Nymphaeaceae). In this genus, we find a gradual transition from leaf-like stamens to stamens that appear more caulomic. According to the transition criterion, it is therefore concluded that all stamens are phyllomes, i.e., they belong to the same morphological category. Stegmann & Schmidt [42] found that psychologically we tend to subsume structures that are linked by transitional forms into the same category. In other words, we conclude that they are essentially the same, although they appear different. However, a transition between different structures shows only that they are related, and “relationship” does not necessarily mean “sameness.” Consider the transition of the following four forms, each of which has three of the traits a to f:

abc

 bcd

  cde

   def

The first and the last form are linked through transitional forms. They are related, but they are not the same. In fact, they have nothing in common. Additionally, even the transitional forms are not the same, although they share traits with the other forms. In terms of homology, this means that they are partially homologous with the others [43]. Minelli [1] referred to factorial and combinatorial homology. Rutishauser and Moline [44] and Minelli [31] discussed yet other homology concepts that have been proposed by various authors. 

Although Stegmann & Schmidt [42] found that we tend to subsume different structures linked by intermediate forms into one category, they also pointed out that “intermediate forms may have the opposite effect, i.e., undermining previous homology claims, while at the same time indicating modified or novel homologies.” This means we need not necessarily succumb to the confusion of sameness and relationship.

## 4. Process Morphology

Many morphologists refer to processes [8,32,33,45]. However, they refer to processes in terms of structures that undergo the processes. This implies a structure/process dualism. In contrast, in process morphology, we transcend the structure/process duality that is characteristic of most biological thinking. This duality implies that there are structures and processes. We discern structures such as plant organs and then we describe the processes that occur within these structures. However, closer inspection reveals that the structures themselves are processes, which means that all is process. For example, a foliage leaf as a functioning part of a living plant is not static. It changes gradually during its life span and thus the leaf itself is dynamic. 

To overcome the structure/process duality, we cannot begin our investigation with structures and then assign processes to these structures. We have to begin with processes. So, what are the most fundamental developmental processes? According to process morphology they are growth and decay, differentiation and dedifferentiation [2,43,46,47,48]. We then distinguish different parameters of these processes. For example, for growth we can distinguish determinate and indeterminate growth, radial and dorsiventral symmetrisation, etc. Additionally, we can quantify the parameters so that they can represent the continuum. Process morphology was first illustrated for the androecium of flowers [49] and two species of *Utricularia* [50].

According to process morphology, the diversity of plant forms represents a diversity of process combinations. The process combinations have been shown to form a continuum, a dynamic continuum [51]. In this continuum, in addition to the process combinations of typical roots, caulomes, and phyllomes, there are process combinations of intermediate forms. The latter combine processes that occur in the typical process combinations. Such combinations have been documented in river-weeds (Podostemaceae), bladderworts (*Utricularia)*, and other flowering plants [1,2,21]. 

River-weeds (Podostemaceae) grow in river-rapids and waterfalls in the tropics. Bell [12] called them “morphological misfits” because they do not fit into the categories of classical morphology. They exhibit developmental processes that combine processes of typical stems and leaves. This has been confirmed by molecular genetics [52].

The genus *Utricularia* (Lentibulariaceae) includes terrestrial, epiphytic and aquatic species. Terrestrial species such as *Utricularia reniformis* may still have process combinations of more typical stems and leaves of flowering plants [24,53,54]. However, among the aquatic species such as *Utricularia aurea, U. australis, U. foliosa, U. gibba*, and *Utricularia dichotoma* (the latter one living in damp or wet habitats in Australia) we find process combinations, often called “stolons” that combine processes of typical stems, leaves, and even roots [2,24,55,56,57]. This is also confirmed through molecular genetic analyses. Based on genomic data, the developmental geneticists Silva et al. [54] (p. 15) presented the following evolutionary hypothesis for two model species: “*Utricularia gibba* seems to have a more severe degree of fuzzy Arberian morphology, such as no clear delimitation of distinct vegetative organs. In contrast, *U. reniformis* presents a more traditional vegetative organ delimitation (as stems and leaves), similar to other angiosperms”. 

## 5. Complementarity

Although classical, continuum and process morphology appear very different or even contradictory, they may be considered complementary [2,24,58]. Classical morphology is less inclusive than continuum and process morphology but can be easily applied to many taxa and in these cases provides a simple description. Continuum morphology is more inclusive than classical morphology, but describing forms that are intermediate between categories is more difficult than categorization. Since plants appear to be profoundly dynamic, process morphology comes closer to reality than classical morphology and continuum morphology based on structure/process dualism. It aims at representing the profoundly dynamic continuum of plant form. However, describing a whole plant or one of its parts only in terms of processes is not an easy task. In this respect, classical morphology and continuum morphology that characterize plant structures in terms of sharp or fuzzy membership are two complementary ways of representation. Therefore, from a practical point of view, classical morphology and continuum morphology that still operate within a structure/process dualism offer useful, though limited, perspectives on plant form. Baum [59] concluded that “depending on the context, parts are best understood sometimes as structures, sometimes as functions, and sometimes as processes.” 

Lacroix et al. [60] considered process morphology the most inclusive approach to the study of plant form and classical morphology a subset of process morphology. Jeune et al. [22] considered both classical morphology and continuum morphology sub-classes of process morphology. However, regardless of how we see the relation between classical, continuum, and process morphology, they cannot always be sharply delimited from one another. Like organ categories, they appear also fuzzy to some extent [2,61]. This is especially evident in classical morphology. When we refer to classical morphology, we mean typical classical morphology. However, more or less deviation from typical classical morphology occurs. This deviation leads to a continuum between classical morphology and continuum morphology [61]. Whereas typical classical morphology is opposed to continuum morphology, classical morphology with fuzzy edges results in a continuum between classical and continuum morphology. Additionally, when continuum morphology transcends the structure/process duality it merges with process morphology in a dynamic continuum of process combinations. 

## 6. Processes in Plant Evo-Devo (Evolutionary Developmental Biology)

We are concerned here primarily with the morphological aspects of plant evo-devo, that is, MorphoEvoDevo, which investigates how structures or process combinations change during plant development and evolution, ontogeny and phylogeny. According to Walter Zimmermann, the founder of the telome theory, there are three fundamental processes: heteromorphy, heterotopy, and heterochrony [39]. Heteromorphy comprises many processes such as homeosis, developmental hybridization, transference of function, and the elementary processes of the telome theory (see below). 

Classical morphology of vascular plants such as flowering plants operates within the trinity of three mutually exclusive organ categories (root, caulome, and phyllome). Through which processes did the three kinds of organs evolve in the first place? Different theories have been proposed. According to Zimmermann’s telome theory the three kinds of organs evolved from radially symmetrical telome trusses (such as those of *Cooksonia*) through the elementary processes of overtopping, planation, and fusion (webbing) [39,62,63,64,65,66,67,68,69]. Overtopping led to the formation of the main axis of the root and stem, planation involved the orientation of telomes into one plane in which they formed a blade as a result of webbing. The latter process may have to be conceptualized differently. Already Zimmermann [39,40] pointed out that at least in part it involves a basipetal shifting (“Verschiebung”) of growth. It seems that shifting of growth (growing zones) constitutes one of the general processes in the development and evolution of the diversity of plant morphology that will have to be investigated in more detail. Furthermore, extensions and novel additions of growing zones such as intercalary meristems have to be considered [70]. 

As an alternative to Zimmermann’s telome theory, Hagemann [71] proposed that vascular plants with roots, stems (caulomes) and leaves (phyllomes) evolved from dorsiventrally flattened structures. These structures gave rise to leaves (phyllomes), and then stems (caulomes) and roots were added as novel formations. Once these plants with the three kinds of organs had evolved, according to classical morphology, further evolution and diversification have been only a modification of the three kinds of organs. For any unusual organ such as, for example, phylloclades or stolons in *Utricularia*, the question could be only from which kind of organ did it evolve. It could be only a modification of one of the three kinds of organs. Additionally, therefore it remained essentially one of the three. Furthermore, organs retained a fixed position such as leaves subtending branches in their axil. As pointed out above, according to Kaplan, a classical morphologist [9], “organs in plants are defined principally by their topographic-positional relationships” [8] (p. 265). Thus, the homology of any deviant organ is determined primarily by its position [8] (p. 5).

In contrast to classical morphology, according to which the evolution of vascular plants such as flowering plants occurred only through the process of modification of the three fundamental kinds of organs, continuum and process morphology allow additional developmental and evolutionary processes such as homeosis and developmental hybridization. Furthermore, new types of organs could have evolved as, for example, suggested by Bower [72], who thought that roots may have evolved as “a new type of haustoria outgrowth.” Or according to the enation theory, Lycophyte leaves evolved as de novo outgrowths from a stem [73]. 

As pointed out above, homeosis is the replacement of one structure or a trait by another one [35]. In terms of process morphology, we would refer to the replacement of one process combination or one process by another one. As long as the replacement occurs within the same organ category, classical morphology can deal with the situation. However, if a plant organ is replaced by an organ of another category or a trait of another organ category, the tenets of classical morphology are undermined. Then, homeosis becomes an evolutionary process not accounted for in classical plant morphology [35,74].

One special case of homeosis is developmental hybridization [35]. In this case, processes of different categories may be combined. For example, in phylloclades processes of shoots and leaves have been combined. As pointed out above, this combination has been documented morphologically and is also supported by data from molecular genetics. 

Besides unusual structures, also common structures such as compound leaves may have evolved through the process of developmental hybridization. Many morphological studies support this view [11,75,76,77,78]. Based on data from plant molecular genetics, Eckardt and Baum [79] concluded “that compound leaves express both leaf and shoot properties”. 

According to classical morphology, the androecium of flowers consists only of phyllomes, which means that all stamens are leaf homologues. However, the androecium is much more complex (see, for example, [58,80]). It encompasses a continuum of structures ranging from phyllomes to caulomes to branchlets and even branches as in *Ricinus* (Euphorbiaceae) that have been interpreted in different ways [2,81,82]. In this continuum, stamens and stamen fascicles may combine processes of typical phyllomes, caulomes, and branches [49]. Thus, the diversity of androecia cannot be understood simply as a modification of phyllomes.

Besides homeosis, including developmental hybridization, another developmental and evolutionary process that transcends classical morphology is heterotopy, which is the formation of a structure (process combination) in a different position, that is, a positional shifting [39,40]. This process may overlap with homeosis, but if the new structure (process combination) is formed in a novel position it may be considered a separate process. For example, branches such as inflorescences that normally arise in a leaf axil may be formed on a leaf. This position contravenes classical morphology, according to which branches should have an axillary position. Therefore, classical morphologists claim that the branches are congenitally fused with the leaf, that is, they actually arise in the axil and only seem to be formed on the leaf. However, congenital fusion cannot be observed and therefore lacks an empirical basis. What we can observe is that in certain taxa branches such as inflorescences are formed on the adaxial side of a leaf [83]. For example, in *Phyllonoma integerrima* (Phyllonomaceae) the inflorescence primordium is initiated on the leaf [84]. Hence, the normal axillary position has shifted to an epiphyllous one. In *Helwingia japonica* (Helwingiaceae) the inflorescence of the leaf is initiated at the very base of the leaf near or in its axil and then due to an intercalary meristem is translocated onto the leaf [85]. There is no observable fusion but an observable intercalary meristem. Zonal growth in flowers with an inferior ovary is also due to an intercalary meristem [80,86,87]. 

The development and evolution of the gynoecium of flowers also involved the process of heterotopy when the position of the placenta or ovule(s) changed [86]. Recognizing these heterotopic changes has far-reaching consequences for our understanding of the gynoecium. Normally the placenta or ovule(s) is formed on a carpel that, according to classical morphology, is defined as a phyllome that bears and encloses a placenta with ovules or just a single ovule. There are, however, also gynoecia in which the placenta and/or ovule(s) arise in the axil of a gynoecial appendage or on the floral apex. A single axillary ovule occurs in *Illicium lanceolatum* (Schisandraceae) and *Ochna atropurpurea* (Ochnaceae) [88,89]. An ovuliferous branch is formed in the axil of a gynoecial appendage in the atypical gynoecia of *Michelia figo* (Magnoliaceae) [90]. In *Myrica gale* (Myricaceae), *Basella rubra* (Basellaceae) and other taxa the floral apex is transformed into a single ovule [91,92]. In yet other taxa the floral apex forms a placenta that bears ovules [92]. These gynoecia are acarpellate because there are no gynoecial appendages (phyllomes) that bear the ovule(s). 

Instead of distinguishing acarpellate and carpellate gynoecia one could redefine the carpel concept as an appendage that encloses the ovule or placenta but does not always bear them [80] (in their glossary only) [93,94]. Such a redefinition would accommodate the heterotopy of the placenta and/or ovule(s) and would render most acarpellate gynoecia carpellate. However, the phenomenon that the placenta and/or ovule(s) may change their position indicates that they are relatively independent of the phyllomic gynoecial appendages. Even when they are formed on the gynoecial appendage as on a classical carpel, they represent additional structures (process combinations). It seems that based on molecular genetics, Mathews and Kramer [95] came to the same conclusion when they stated that the carpel is a complex lateral organ consisting of a foliaceous appendage (gynoecial appendage) and the placenta and that the latter “has all the hallmarks of axillary meristem,” which indicates its affinity to a shoot. Accordingly, the placenta and/or ovule(s) are not comparable to parts of a phyllome such as leaflets. Rather, they arise from “their own kind of meristematic axis” [95]. 

In addition to heterotopy, heterochrony is also an important developmental and evolutionary process [96]. Heterochrony changes the timing of developmental events. An extreme example of heterochrony occurred in the genus *Balanophora* (Balanophoraceae) [39]. Here, neither gynoecial appendages nor ovules are formed. The floral apex becomes transformed into an elongated structure that internally forms an embryo sac. Hence, the gynoecium is acarpellate even according to the above redefinition of the carpel concept.

Another evolutionary process is the transference of function [97,98]. An example is the transference of the function of the style in flowers to the androecial tube in the genus *Stylidium* (Stylidiaceae) [86,99]. 

## 7. Future Research

1. In studying the development and evolution of vascular plants, continuum and process morphology can inspire future research and direct it into more productive avenues than classical morphology [2]. Instead of seeing the development and evolution of plants such as flowering plants only as a modification of the three basic organ categories of classical morphology, other pathways can be envisaged and have already been demonstrated to some extent. Novel organs may arise as a novel mingling of traits or processes of different structures or process combinations. Even processes of different organ categories may be combined. At the molecular genetic level genes can be co-opted from other process combinations, including those of other organ categories [38,100,101]. Pursuing such investigations can lead to insights not available to classical morphology. They will be aided by an emphasis of process morphology, mathematical analyses, and molecular genetics. They will contribute to research in plant evo-devo, especially MorphoEvoDevo, that is, evolutionary developmental morphology (see also [102]). 

2. We need to further explore the continuum of plant form. This continuum bridges the mutually exclusive categories such as root, stem, and leaf of classical morphology. As pointed out above, intermediate forms that do not fit into the classical categories have been called misfits. “Being a misfit is not the problem of the plant, but the problem of our inadequate thinking and concepts” [2]. We need to pay more attention to these so-called misfits and investigate how they developed and evolved. They cannot be understood in terms of homology as one-to-one correspondence, but require concepts of partial homology, factorial and combinatorial homology. In this regard, we have to pay attention to the partial iteration of structural units or process combinations such as, for example, the partial iteration of shoot processes in compound leaves. These findings are relevant to evo-devo, especially MorphoEvoDevo.

3. We need to appreciate and further explore the complexity of plant form. “We should not confuse our favourite metaphors such as root, shoot, leaf and flower with reality, which is more complex” [103]. The interaction of continuum and process morphology with molecular genetics in evo-devo research can lead to novel insights that may reveal the complexity of plants beyond the simplistic categorical framework of classical morphology. Patterns of gene expression are complex. Morphological diversity is complex. Additionally, the relationship between morphology and molecular genetics is complex (e.g., [2,100]). 

4. In addition to the common Aristotelian identity and either/or logic, we have to use fuzzy logic and both/and logic to better understand the complexity of the diversity of plant form [2,24]. This may require changing the logic of the questions we ask. We cannot take it for granted that it is meaningful or appropriate to ask whether a particular structure or process combination is either this or that, for example, either a stem or a leaf. Additionally, we cannot take it for granted that questions about the identity of a structure or process combination are always meaningful. If we do not find satisfactory answers to our questions it may be because the questions we ask imply a logic that is inappropriate for the particular situations. 

5. A reconciliation and synthesis of different or even contradictory theories and paradigms appear highly desirable. Continuum and process morphology may have this potential. For example, as pointed out above, there are two contradictory theories of the origin of vascular plants. According to the telome theory [39,62], vascular plants originated from radially symmetrical telome trusses, whereas according to Hagemann’s theory [71], they arose from dorsiventrally flattened structures. Recognizing a continuum between radial and dorsiventral symmetry, for which there is also some fossil evidence, could reconcile the two theories [104]. Another example: A reconciliation and synthesis of classical morphology and continuum morphology could be achieved through the notion of the extreme type. The extreme type has fuzzy edges and through these fuzzy edges different extreme types such as those of stem and leaf become continuous. Contrary to the extreme type, the classificatory type that is normally used in classical morphology has sharply defined boundaries according to Aristotelian identity and either/or logic. If classical morphologists would use the notion of the extreme type instead of the classificatory type concept, they would become continuum morphologists and yet they could retain their typological framework. Classical and continuum morphology would cease to be in opposition [61]. 

6. Ideas and theories that cannot be reconciled may be considered complementary perspectives at least in some cases [2,24,58]. Complementary perspectives present a more inclusive picture of reality than only one of these perspectives. Thus, instead of fighting the author who defends a different or even contradictory theory, one can embrace him or her, if not physically, at least symbolically. Let us follow the lead of the quantum physicist Niels Bohr who recognized the complementarity of the particle and wave views of light. 

7. Finally, we need not only intellectual analysis but also a “feeling of the organism” [105] and a recognition of the complementarity of science and the arts, the scientific exploration of plant form and its aesthetic appreciation. Kirchoff [106] suggested that we need a holistic aesthetic based on love. In loving absorption with plant forms and all of nature we may even transcend our ordinary perception of space and time (hence we even transcend process) as beautifully expressed by William Blake: 

To see the world in a grain of sand,

And a heaven in a wild flower,

Hold infinity in the palm of your hand,

And eternity in an hour.

## Data Availability

Not applicable.

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
