# Peer review of "Fundamentals of Plant Morphology and Plant Evo-Devo (Evolutionary Developmental Morphology)"

_plants, 2022, doi:10.3390/plants12010118_

Round 1

Reviewer 1 Report

In this review, Sattler and Rutishauser discuss the fundamentals of plant morphology and its evolutionary development. "Classical", "continuum", and "process" views on plant morphology are presented and discussed in this interesting essay. 

My comments are minor, mostly regarding suggestions to improve the clarity and fluidity of the text. 

Author Response

We considered all suggestions made by the three reviewers and changed the manuscript accordingly. We also incorporated all the new references suggested by the reviewers. More specifically, we made all the minor changes suggested by the first reviewer.

Reviewer 2 Report

Sattler and Rutishauser wrote an interesting essay on the topic of Plant Morphology and Plant Evo-Devo. I thought the essay was well written and stimulating and will be of interest to a range of plant scientists. I also thought the essay is written in an accessible way that will make it of interest for students. I have 4 general comments and some specific comments:

1.      I think the authors could potentially highlight in the essay how angiosperm centric classic morphology has been and how this adds challenges when trying to use these concepts in an evolutionary approach across all land plants.

2.      The section on Classical Morphology, might benefit from highlighting that Goethe’s work, and therefore much of classic morphology was built on pre-Darwinian foundations. In this section you could also mentioned slightly later authors we contributed to discussing the earlier concepts of Goethe in an evolutionary framework such as F. O. Bower.

3.      I think the section on evolutionary theories such as the telome theory could be extended. There have been quite a large number of criticisms and limitations of the telome theory that could be mentioned. For example, https://www.jstor.org/stable/4096961 and https://doi.org/10.1086/513519. There are large numbers of fossil discoveries providing evidence for the structure of early land plants described since Zimmerman’s work. For example those discussed in Kenrick P, Crane PR. 1997. The Origin and Early Diversification of Land Plants: A Cladistic Study. Smithsonian Series in Comparative Evolutionary Biology. Washington, DC, USA: Smithsonian Institute Press.  Finally, I am unaware of any additional evidence put forward in support of Hagemann’s theory – if there has been more discussion of it or fossil evidence supporting the theory it may be useful to discuss.

Specific

Line 145: ““It is  now widely accepted that … radiality and dorsiventrality are but extremes of a continu-146 ous spectrum. In fact, it is simply the timing of the KNOX gene expression.”” This seems quite over simplistic as KNOX genes have a plethora of functions with evolutionary origins predating land plants. E.g. https://doi.org/10.1242/dev.030049 and https://doi.org/10.1111/nph.14318

Line 150: “the position of an organ determines its homology”. This discussion would benefit from slightly more introduction to homology. Especially for many comparative evolutionary approaches taxic homology (Patterson C. 1982. Morphological characters and homology. In Joysey KA, Friday AE. ed; Problems of Phylogenetic Reconstruction. London: Academic Press. p 21–74) can be the most useful. This is particularly relevant for the current essay as it switches the focus away from defining structures relative to three fundamental kinds of organs and instead means that structure should be defined and interpreted within a specific clade. This is particularly important as there is evidence that leaves and roots had multiple independent origins in vascular plants (e.g. https://doi.org/10.1016/j.pbi.2009.10.001).

Line 299-300: “it could be only a modification of one of the three kinds of organs”. Can’t it also be considered a new type of organ. E.g. when describing the origin of roots, Bower 1908 (Bower FO. 1908. The Origin of a Land Flora. Macmillan and Co.). States that roots may have evolved as “a new type of haustorial outgrowth, not originally of shoot-nature” Pg. 221.  Or when considering the enation theory for the origin of lycophytes leaves. Leaves are predicted to have evolved as de novo outgrowths from a stem. E.g. (Kenrick, P., 2002. The telome theory. Developmental genetics and plant evolution, pp.365-387.)

Line 449-466: I am not sure these final couple of paragraphs add anything to the essay.

Author Response

We considered all suggestions made by the three reviewers and changed the manuscript accordingly. We also incorporated all the new references suggested by the reviewers. Concerning the second reviewer, we added a paragraph in which we pointed out that Goethe’s approach was pre-Darwinian and angiosperm-centric, we added references that were critical of the telome theory, we added a sentence indicating that Knox genes have also a plethora of other functions predating land plants, we elaborated briefly on homology, and we indicated that new types of organs may be formed as suggested by the reviewer. We retained the final couple of paragraphs because we are convinced that they add important dimensions to our paper. 

Reviewer 3 Report

The subject of this review is familiar to me from what I know of the authors' work over the past half century (in the case of Rolf Sattler). I am sympathetic to the authors' argument and believe that it is important to have its exposition as part of this Special Issue. Sattler's critical review of Kaplan's posthumous publication, Principles of Plant Morphology (ed. C.D. Specht, 2022) is helpful, but reactive. This review provides an independent statement of a point of view that in my opinion is best suited to analytically linking descriptions of plant morphology to the abundant new molecular data on the control of plant development, and doing so in a context compatible with phylogenetic analyses.

Regarding the importance of process, the authors appear to neglect the way in which variation in morphogenetic processes were (perhaps unintentionally) emphasized in an "Illustrated Key to the Architectural Models of Tropical Trees," found on pp. 84-97 of Tropical Trees and Forests: An Architectural Analysis by F. Halle, R.A.A. Oldeman, and P.B. Tomlinson (1978). I say, perhaps unintentionally, because Tomlinson presented a late draft of this key to students in a field course on tropical plants. Tomlinson seemed to strenuously resist the students' desire to see this key not only as an identification tool but also as the basis for a possible flowchart of branch points leading to different tree architectures mediated by modifications of morphogenetic processes (rhythmicity, dominance, flowering, etc.) and their spatial configuration. That resistance, if that is what it was, seems a pity in hindsight because embracing the students' perspective could have transformed the concept of purely descriptive models into an overarching, testable model for the evolution of tree architectures.

Author Response

We considered all suggestions made by the three reviewers and changed the manuscript accordingly. We also incorporated all the new references suggested by the reviewers. More specifically, we incorporated the reference suggested by the third reviewer.